# Post-discharge tobacco abstinence in a Mumbai hospital after implementation of tobacco cessation counseling: A pragmatic evaluation of the LifeFirst program

**Himanshu A. Gupte**[1⊚], **Gina R. Kruse**[2⊚]*, **Yuchiao Chang**[3,4], **Dinesh Jagiasi**[1], **Sultan Pradhan**[5], **Nancy A. Rigotti**[3,4]

1 Narotam Sekhsaria Foundation, Mumbai, India, 2 Division of General Internal Medicine, University of Colorado, Denver, Colorado, United States of America, 3 Division of General Internal Medicine, Tobacco Research and Treatment Center, Massachusetts General Hospital, Boston, Massachusetts, United States of America, 4 Harvard Medical School, Boston, Massachusetts, United States of America, 5 Prince Aly Khan Hospital, Mumbai, India

⊚ These authors contributed equally to this work.
* gina.kruse@cuanschutz.edu

**Data Availability Statement:** All relevant data are within the paper and its Supporting Information files.

## Abstract

### Background

Hospitalization provides a key opportunity to address tobacco use. Few studies have examined cessation treatment in hospitals in low- and middle-income countries (LMIC). We aimed to measure tobacco abstinence among individuals discharged from a Mumbai hospital after the implementation of cessation counseling compared to abstinence among those discharged pre-implementation.

### Methods

Pre-post intervention study in the Prince Aly Khan Hospital, Mumbai pre- (11/2015-10/2016) and post-implementation (02/2018-02/2020) of LifeFirst counseling. LifeFirst is multi-session (up to six sessions) counseling extending from hospitalization up to six months post-discharge. Primary analyses compare self-reported 6-month continuous abstinence among hospitalized individuals post-implementation (intervention) with pre-implementation (comparator) using an intent-to-treat approach that includes all participants offered LifeFirst post-implementation in the intervention group. Secondary analyses compare those who received ≥ 1 LifeFirst session with the pre-implementation group.

### Results

We enrolled n = 437 individuals pre-implementation (8.7% dual use, 57.7% smokeless tobacco, 33.6% smoking) and n = 561 post-implementation (8.6% dual use, 64.3% smokeless tobacco, 27.1% smoking). Post-implementation, 490 patients (87.3%) accepted ≥ 1 counseling session. Continuous abstinence 6-months post-discharge was higher post-implementation (post: 41.6% vs. pre: 20.0%; adjusted odds ratio [aOR]: 2.86, 95%

**Funding:** This project was funded by the Narotam Sekhsaria Foundation. HG is an employee of the Narotam Sekhsaria Foundation and as a lead author, participated in study design, oversaw data collection, collaborated on the analysis and interpretation of data, the writing of the report and in the decision to submit the paper for publication. The funder had no other role in study design, data collection analysis or interpretation, writing of the report or the decision to submit.

**Competing interests:** GK has a family financial interest in Dimagi Inc. She was not involved in selection of the data collection platform. GK has received research grants from the National Comprehensive Cancer Network with funding from Astra Zeneca. HG is an employee of the Narotam Sekhsaria Foundation. All other authors declare that they have no conflicts of interest. This does not alter our adherence to PLOS ONE policies on sharing data and materials

confidence interval [CI] 1.94–4.21). Those who received LifeFirst had higher odds of continuous abstinence compared to pre-implementation (aOR: 2.95, 95% CI 1.98–4.40).

## Conclusion

Post-discharge abstinence was more common after implementation of a multi-session tobacco counseling program for hospitalized patients compared to abstinence among patients hospitalized before implementation. These findings represent observational evidence of a promising association between post-discharge abstinence and a hospital-based tobacco cessation program implemented within routine practice in an LMIC setting.

## Introduction

Hospitalization provides an important context and an opportune time to address tobacco use. Providing cessation assistance at the bedside can support the initiation of a quit attempt, and continued support after discharge increases the likelihood of long-term tobacco abstinence. For individuals who are hospitalized for a tobacco-related illness, cessation is associated with improved outcomes including improved treatment response and survival among patients with head and neck cancer [1], reduced likelihood of developing secondary cancer after treatment [2], and reduced mortality after myocardial infarction [3]. There is also evidence that hospital tobacco treatment services are associated with reduced 30-day readmissions and healthcare costs post-discharge [4, 5].

Cochrane meta-analyses of smoking cessation treatment for hospitalized patients reveal that counseling alone significantly increases the likelihood of cessation at 6 months after discharge if extended for at least one month after hospitalization (risk ratio [RR] 1.36; 95% confidence interval [CI] 1.24–1.49) [6]. However, most of the evidence supporting the effectiveness of hospital-initiated cessation services was produced in high-income countries [7]. The Cochrane systematic review included 82 hospital-initiated smoking cessation intervention studies (74 randomized controlled trials) but only three trials were conducted in LMICs (Brazil, Tunisia, China) [6]. A recent review of randomized controlled tobacco cessation trials identified 92 trials conducted in LMICs, 24 of which included long-term outcomes, and of these, only two (the study conducted in a Brazilian hospital and an inpatient study in China) examined hospital-based treatment [8–10]. Indeed, the availability of hospital-based tobacco treatment services is limited in low- and middle-income countries (LMICs) where over 80% of tobacco-attributable deaths occur [11, 12]. In a 2017 survey, only about half of LMICs surveyed reported cessation services were available in hospitals [13]. In India, the absence of tobacco cessation services in secondary and tertiary care and private healthcare is highlighted as a challenge in the search for cessation solutions [14]. There is clearly a pressing need for practice-based evidence from LMIC settings.

Another gap in the evidence supporting hospital-initiated cessation interventions is the effectiveness of treatments for smokeless or other forms of tobacco other than manufactured cigarettes [15]. Research on cessation of smokeless tobacco or dual forms of tobacco in any setting remains limited [16–18]. Smokeless tobacco produces morbidity and mortality from cancer and cardiovascular disease and this burden falls primarily in LMICs [19–21]. India, for example, is home to 267 million people who report current tobacco use [22]. This equates to 42% of Indian males over 15 years of age, of which 19% smoke and 30% use smokeless tobacco [23]. Among Indian women over age 15 years, 13% use smokeless tobacco and 2% smoke. The

effects of this tobacco use are apparent in tobacco related disease incidence and prevalence. In 2020, India had an estimated 135,929 new cases of lip and oral cavity cancer, making it the second most common cancer in India after breast (178,361) [24].

Given this urgent need to study hospital-based tobacco cessation treatments in real-world LMIC settings including support for smokeless and other forms of tobacco, we designed a study to evaluate a hospital-based tobacco cessation counseling program that was developed and implemented in India.

## Methods

Using a pre-post design, we measured the prevalence of tobacco abstinence after discharge among patients hospitalized in Mumbai before and after implementing a multisession tobacco cessation counseling program.

### Sample

The study sample includes patients admitted to the Prince Aly Khan Hospital, a 137-bed private general hospital in Mumbai, India, before (11/01/2015-10/31/2016) and after (02/01/2018-02/28/2020) the implementation in 2017 of a tobacco cessation counseling program. In both the pre- and post-implementation periods, all patients admitted to the hospital were identified daily from the hospital management information system admission report. Using the same instruments and recruitment methods for both periods, research staff approached each newly admitted patient to assess their eligibility and invite them to participate in a study. For pre- and post-implementation patients who were eligible and interested in the study an in-person baseline survey was conducted along with hospital medical record review. Patients who reported current tobacco use in the survey were offered brief advice comprised of an in-hospital tobacco cessation assessment and advice to quit [25, 26]. Post-implementation, participants were offered the LifeFirst counseling program comprised of one in-hospital session and up to six telephonic counselling sessions. Both pre- and post-implementation participants were contacted by telephone four times over six months post-discharge using the same instruments to ascertain cessation outcomes.

Inclusion criteria included being an Indian National or Indian resident, over 15 years of age, admitted for ≥24 hours, speaking Hindi, Marathi, or English and reporting current tobacco use. Current tobacco use was defined as an "Every day" or "Some days" response to the Global Adult Tobacco Survey questions "Do you currently smoke on a daily basis, some days or not at all?" and/or "Do you currently use smokeless tobacco on a daily basis, some days, or not at all?" [27]. Patients were excluded for medical instability, cognitive impairment limiting their ability to consent or participate in study activities, or admission to Intensive Care Units, Day Care Units, or High Dependency Units.

Written informed consent to participate was obtained in participants' preferred language prior to enrollment. In the case of minors (i.e., 15-17-year-olds), written informed consent from a parent or guardian plus written assent from the minor were obtained prior to taking part in the research.

### LifeFirst intervention

In the post-implementation period, all admitted patients who reported current tobacco use were offered LifeFirst. LifeFirst is comprised of one in-person session in the hospital and up to six post-discharge telephone counseling sessions at 7 days, 15 days, 1 month, 2 months, 4 months, and 6 months post-discharge.

The program content is guided by the Transtheoretical Model of Change [28]. It includes education on the health effects of tobacco and benefits of cessation, motivational content to promote self-efficacy and to move participants from contemplation or preparation to take action with quit attempts after discharge, assessment of readiness to quit, assessment of past quit attempts, triggers and nicotine dependence, education about cessation treatment options, encouragement to set a quit date, education about withdrawal symptoms, dealing with cravings, engagement of social support, and managing stress (See S1 Table). Pharmacotherapy is not integral to the program due to the high costs of nicotine replacement therapy and prescription pharmacotherapies and because of the large proportion of patients using smokeless only, for whom there is less evidence supporting pharmacotherapy use. The in-hospital session lasts from 30–45 minutes and telephonic follow-up sessions last for 15–20 minutes each. Content does not include pharmacotherapy counseling or recommendations.

Counseling is delivered by staff trained in counselling and behavior modification techniques including motivational interviewing. LifeFirst training for counsellors consists of 12 hours of in-person tobacco cessation counselling training using a standardized manual. All counselors completed training prior to the post-implementation period. Counsellors additionally participated in refresher trainings in-person every six months throughout the post-implementation period. During the study, there were three LifeFirst Counselors. One counselor had a Masters level degree in Clinical Psychology and two with Masters in Social Work with specialization in Counselling.

To maintain uniformity in the intervention across counselors the LifeFirst content was delivered using a guide comprised of a questionnaire and corresponding counseling topics. To promote intervention fidelity, LifeFirst program uses peer observation every 3–6 months in a subset of sessions. This observation includes a checklist aligned with the LifeFirst guide. Peers then provide feedback based on the checklist findings. Counseling is delivered in Hindi, Marathi, or English according to participant preferences.

## Assessments

Survey items were developed in English, translated and back-translated into Hindi and Marathi. Translations and back-translations were then reviewed by staff fluent in these languages. Any edits or clarifications based on this review were discussed and agreed upon by the study team. The same instruments were used in pre- and post-implementation groups (S1 File). The baseline assessment was an in-person survey conducted in the hospital at the bedside or in a designated exam room. Research staff used electronic tablets equipped with a digital data collection and management platform (CommCare) [29]. The survey included measures of educational attainment, employment status, marital status, tobacco products used, cessation interventions used, Fagerström test for nicotine dependence and Fagerström test for nicotine dependence-smokeless tobacco version [30, 31], measures of confidence and motivation to quit tobacco that use a four-item Likert scale response structure (not confident/somewhat confident/very confident/I don't know; and not motivated/somewhat motivated/very motivated/I don't know), and health beliefs about the harms of tobacco ("Do you think tobacco has harmed your health?" with response options of not at all/a little bit/some/a lot/I don't know). Data extracted from the medical record included age, gender, admitting department, admission diagnosis, and admission date. Participants were categorized as currently smoking, currently using smokeless tobacco or currently using both product types. Admission diagnoses were categorized as: cardiovascular disease, tobacco-related cancer (bladder, cervical, colorectal, esophageal, renal, laryngeal, acute leukemia, liver, lung, oral, pharyngeal, pancreatic, and stomach) [32], non-tobacco related cancer, chronic lung disease, or other diagnoses by a physician

investigator trained in Internal Medicine (GRK). Participants were asked for a contact number for themselves and for a proxy who could report on participants' tobacco use in follow-up telephone assessments.

Follow-up surveys were conducted by research staff not involved in delivering the LifeFirst counseling intervention. Surveys were conducted by telephone at 1-week, 1-, 3- and 6-months after hospital discharge. If an individual was not reached on their follow-up date, study staff attempted contact up to four times for up to seven days after the seven-day follow-up, up to 15 days for the 1-month follow-up, up to 30 days for the 3-month follow-up and up to 60 days for the 6-month follow-up. At each follow-up tobacco use was assessed with the question "Since leaving the hospital, have you used any [tobacco product]?". Cessation intervention use was assessed with "Yes" or "No" responses to the item "Since leaving the hospital, did you use any of the following to try to stop [smoking/using smokeless tobacco]?".

## Outcomes

The primary outcome was self-reported continuous tobacco abstinence for 6-months after hospital discharge. Secondary outcomes included continuous abstinence from discharge to the other follow-up points (1-week, 1- and 3-months). To be considered continuously abstinent at each follow-up point, the participant had to report continuous abstinence from date of discharge to that assessment and not have previously reported having used tobacco at any prior follow-up assessment. If participant or proxy responded *YES* to using any tobacco since leaving the hospital, all subsequent follow-ups were coded with a *YES* response. If a follow-up was missed and the respondent reported continuous abstinence at any of the subsequent time points then missing outcome status was inferred as continuous abstinence. For example, if a participant missed the 3-month follow-up and the patient reported continuous abstinence from smoking at the 6-month follow-up then the patient's 3-month status was coded as continuously abstinent. When participant or proxy report was not available and tobacco use outcome cannot be deduced from subsequent time points, the tobacco use outcome at each follow-up was imputed using multiple imputation as described below.

## Sample size and power

Sample size calculations in this study were sequential. First, the pre-implementation cohort size of n = 437 was based on achieving 3% margin of error in an inpatient sample with an estimated 17% current tobacco use prevalence based on summary hospital data. Our power calculation for the pre-post comparison was made using data from the pre-implementation sample showing a prevalence of self-reported continuous abstinence of 20.6%. We selected the post-implementation sample size to have sufficient power to detect a difference in abstinence at 6 months compared to the pre-implementation cohort. With a total cohort size of n = 998 and alpha set at 5%, we had greater than 80% power to detect an increase in continuous abstinence from 20.6%, as observed in the pre-implementation group, to 29.2% in the post-implementation group. This increase is a relative risk of 1.37, the effect size calculated in the Cochrane review of cessation interventions for hospitalized patients [33].

## Statistical analysis

We compared descriptive data for the pre- and post-implementation samples using chi-square for categorical variables and two-sample t-tests or Wilcoxon rank sum tests for continuous variables. For participants missing any follow-up responses, we used a series of sequential multiple imputation models due to the cumulative nature of the outcome assessed over time. We started with imputing outcome at 1 week, followed by outcome at 1 month, 3 months, then

outcome at 6 months. If missing outcome was imputed as non-continuous abstinence, all subsequent follow-ups were imputed as non-continuous abstinence. We opted for multiple imputation rather than a missing = smoking approach because missing = smoking is highly sensitive to differences in rates of missing outcomes between groups [34]. The sequential multiple imputation model included period (pre- or post-), type of tobacco used at baseline, age, gender, education, employment status, marital status, admission diagnosis, FTND score, previous pattern of smokeless and smoked tobacco, intervention from health care providers, baseline depression and anxiety symptoms, and number of counseling sessions previously received. Using the imputed samples and logistical regression models, we compared the unadjusted odds of continuous abstinence from all tobacco 6-months after hospital discharge in post- versus pre-implementation samples and the odds of continuous abstinence adjusted for demographic, socioeconomic, and tobacco use characteristics that differed between the two samples. We conducted several sensitivity analyses examining the effect (1) limited to the subgroup with self-reported or proxy-reported surveys, excluding imputed outcomes (2) limited to only the subgroup with self-reported surveys, excluding imputed or proxy-reported outcomes. Secondary analysis examined continuous abstinence among people who received one or more Life-First sessions in follow-up compared to participants in both groups who received no sessions. Test statistics with p<0.05 were considered significant. All analyses were conducted using SAS version 9.4 (Cary, NC). NC). The study was reviewed and approved by the Prince Aly Khan Ethics Committee, the Mass General Brigham Institutional Review Board, and the Indian Council on Medical Research. Informed consent was obtained from all individuals included in the study.

## Results

The flow of participants through screening and enrollment is shown in Fig 1. Characteristics of patients at baseline in the pre- and post-implementation samples are shown in Table 1. Pre- and post-implementation groups did not differ significantly in demographic factors (age, sex, education, employment, or marital status), admitting diagnosis, or type of tobacco product. Tobacco smoking only was reported by 33.6% of participants pre- vs 27.1% of participants post-implementation, smokeless tobacco only by 57.7% (pre) vs. 64.3% (post), and 8.7% (pre) vs 8.6% (post) for dual tobacco use (p = 0.07) (Table 1). Specific products used are shown in S2 Table.

Pre- and post-implementation groups differed in measures of nicotine dependence and symptoms of depression and anxiety. The proportion of individuals who reported that they were very confident in their ability to quit and that they were very motivated to quit also differed, being higher post-implementation compared to pre-implementation (all p < .001). At baseline, post-implementation participants also more often reported a belief that tobacco has harmed their health (p<0.001).

Use of pharmacotherapy and evidence-based behavioral treatment for smoking were rarely reported at baseline. Nicotine replacement therapies were used in a past year quit attempt by 2.1% (n = 9) pre-implementation and 1.4% post (n = 8). Use of bupropion or varenicline in a past year quit attempt was reported by 0.5% (n = 2) pre-implementation and 0.2% post (n = 1). Use of counseling or a quitline was reported at baseline by 0.9% (n = 4) pre-implementation and no participants at baseline post-implementation.

In the post-implementation period, 490 (87.3%) accepted one or more LifeFirst follow-up counseling sessions. The mean number of follow-up sessions received by LifeFirst participants was 4.64 (standard deviation = 1.25). In both pre- and post-implementation, most assessments were completed by patients rather than proxies. The proportion of assessments completed by

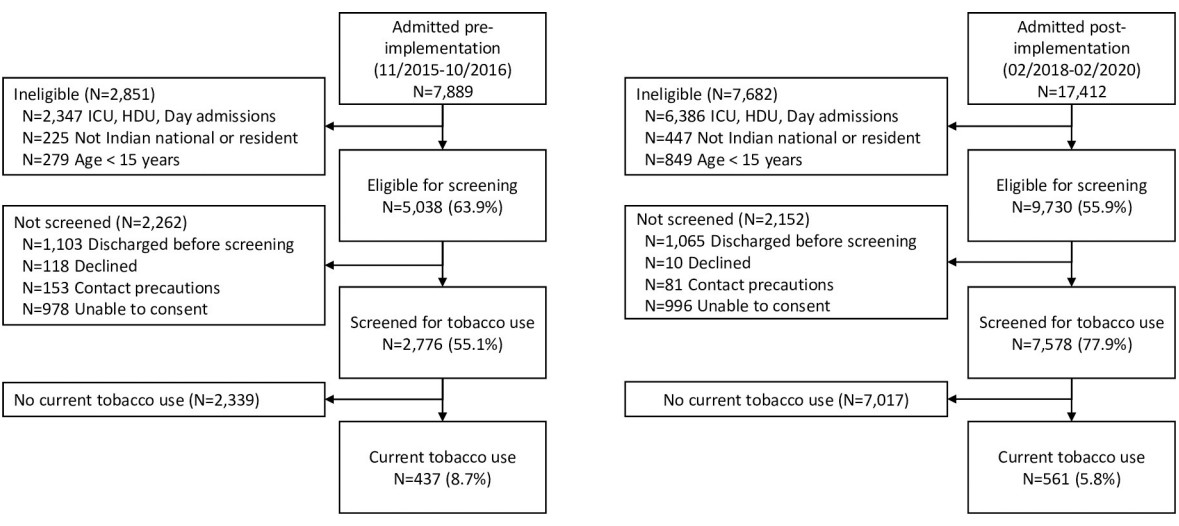

**Fig 1. Flow of participants.**

self or proxy was higher post-implementation at all follow-up timepoints (S3 Table; 1 week: pre 62.5% vs. post 83.1%; 1 month: pre 58.1% vs. post 80.2%; 3 months: pre 51.7% vs. post 70.6%, 6 months: pre 49.7% vs. post 70.4%, differences all p<0.001). Proportion of self-reported outcome assessments was higher post-implementation with the exception of at the 6-month follow-up for the primary outcome, when there was no significant difference in self-reported assessment (82.9% of 6-month responses were self-report pre-implementation vs. 87.1% post-implementation, p = 0.16).

Self- or proxy-reported cessation in the pre- and post-implementation samples from all forms of tobacco and stratified by tobacco product type are displayed in Fig 2. While abstinence was similar in both groups at 1-week post-discharge, the proportion reporting continuous abstinence began to decline more quickly in the pre-implementation group starting at 1-month. Continuous abstinence from all forms of tobacco at 6-months post-discharge was reported by 20.0% pre-implementation compared to 41.6% post-implementation.

The unadjusted odds of continuous abstinence from all tobacco at 6-months were higher post-implementation compared to pre-implementation (odds ratio [OR] 2.86, 95% confidence interval [CL] 2.06–3.96, p<0.0001) (Table 2). In models adjusting for demographics, socioeconomic characteristics, nicotine dependence, psychosocial measures and health beliefs, there were higher odds of continuous abstinence at 6-months post-implementation compared to pre-implementation (adjusted OR [aOR] 2.86, 95% CI 1.94–4.21, p<0.001) (Table 2). The only characteristic other than pre- or post-implementation phase that was significantly, independently associated with continuous abstinence in the adjusted model was tobacco-related admission diagnosis (aOR 2.36, 95% CI 1.71–3.26). In sensitivity analyses excluding imputed outcomes, the adjusted odds of continuous abstinence from all tobacco at 6-months post-discharge were significantly higher in the post-implementation sample compared to pre-implementation (aOR 2.76, 95%CI 1.72–4.44; p<0.001, S4 Table). If further limited to self-reported outcomes only, the adjusted odds of continuous abstinence from all tobacco at 6-months post-discharge were significantly higher in the post-implementation sample compared to pre-implementation (aOR 2.36, 95%CI 1.31–4.23; p = 0.004, S4 Table).

Secondary analyses compared only participants who accepted the LifeFirst intervention post-implementation (n = 490) with pre-implementation participants (n = 437). Continuous abstinence from all forms of tobacco at 6-months post-discharge among those who accepted

**Table 1. Characteristics of hospitalized current tobacco users in Mumbai, n = 998.**

| | Pre-implementation of LifeFirst | | Post-implementation of LifeFirst | | p-value[1] |
|---|---|---|---|---|---|
| | 11/2015-10/2016 | | 02/2018-02/2020 | | |
| | N | % | N | % | |
| **TOTAL** | 437 | 100.0 | 561 | 100.0 | |
| Baseline current tobacco use | | | | | 0.07 |
| Dual tobacco | 38 | 8.7 | 48 | 8.6 | |
| Smokeless only | 252 | 57.7 | 361 | 64.3 | |
| Smoking only | 147 | 33.6 | 152 | 27.1 | |
| **DEMOGRAPHIC/SOCIOECONOMIC** | | | | | |
| Age in years-median (IQR) | 52.0 | 40.0–61.0 | 51.0 | 40.0–60.0 | 0.81 |
| Female | 94 | 22.1 | 116 | 20.9 | 0.66 |
| Level of education | | | | | 0.17 |
| Illiterate | 45 | 10.3 | 50 | 8.9 | |
| No formal schooling, can read | 35 | 8.0 | 28 | 5.0 | |
| Primary school | 146 | 33.4 | 215 | 38.3 | |
| Secondary school or more | 211 | 48.3 | 267 | 47.6 | |
| Don't know/refused | – | – | 1 | 0.2 | |
| Employment status | | | | | 0.31 |
| Employed | 128 | 29.3 | 157 | 28.0 | |
| Self-employed | 147 | 33.6 | 219 | 39.0 | |
| Not working | 80 | 18.3 | 86 | 15.3 | |
| Homemaker | 82 | 18.8 | 99 | 17.6 | |
| Married | 369 | 84.4 | 462 | 82.4 | 0.38 |
| **TOBACCO CHARACTERISTICS** | | | | | |
| FTND[2]-mean (SD) | 4.2 | 2.5 | 3.6 | 2.3 | <0.001 |
| Confidence that you could stop tobacco after leaving the hospital | | | | | <0.001 |
| Not confident | 60 | 13.7 | 61 | 10.9 | |
| Somewhat confident | 166 | 38.0 | 168 | 29.9 | |
| Very confident | 196 | 44.9 | 326 | 58.1 | |
| Don't know/Refused | 15 | 3.4 | 6 | 1.1 | |
| Motivation to quit after leaving the hospital | | | | | <0.001 |
| Not motivated | 60 | 13.7 | 80 | 14.3 | |
| Somewhat motivated | 176 | 40.3 | 182 | 32.4 | |
| Very motivated | 185 | 42.3 | 293 | 52.2 | |
| Don't know/Refused | 15 | 3.4 | 6 | 1.1 | |
| Belief in harms of tobacco[3] | | | | | <0.001 |
| Most | 97 | 22.2 | 147 | 26.2 | |
| Some | 163 | 37.3 | 382 | 68.1 | |
| Little | 177 | 40.5 | 32 | 5.7 | |
| **HEALTH STATUS** | | | | | |
| Admission diagnosis | | | | | 0.19 |
| Cardiovascular disease | 59 | 13.5 | 73 | 13.0 | |
| Tobacco-related cancer[4] | 87 | 19.9 | 144 | 25.7 | |
| Non-tobacco related cancer | 37 | 8.5 | 35 | 6.2 | |
| Chronic lung disease | 3 | 0.7 | 6 | 1.1 | |
| Other diagnoses | 251 | 57.4 | 303 | 54.0 | |
| Depression symptoms-PHQ-2 | 92 | 21.1 | 46 | 8.2 | <0.001 |

(*Continued*)

**Table 1.** (Continued)

|  | Pre-implementation of LifeFirst | | Post-implementation of LifeFirst | | p-value[1] |
|---|---|---|---|---|---|
| Anxiety symptoms-GAD-2 | 87 | 19.9 | 62 | 11.1 | <0.001 |

[1] Based on Pearson's chi-square for categorical variables and t-tests for continuous variables.; [2] Fagerström test for nicotine dependence and Fagerström test for nicotine dependence-smokeless tobacco version. For dual users, the highest score of the two is included.; [3] Composite item of "Do you think tobacco use has harmed your health", "Is tobacco a cause of the illness you are in the hospital for?" and "Do you think quitting now would improve your health?" with responses categorized as Most: reported *Some* or *A lot* for all three items; Some: reported at least one *Some* or *A lot* in any of the three items; Little: reported *A little bit*, *Not at all*, or *Don't know/refused* for all three items.; [4] Based on 2014 U.S. Surgeon General's Report, includes bladder, cervical, colorectal, esophageal, renal, laryngeal, acute leukemia, liver, lung, oral, pharyngeal, pancreatic, and stomach.

LifeFirst was 42.7%. The odds of cessation were higher in those who accepted LifeFirst counseling compared to the pre-implementation participants (OR 3.00, 95% CI 2.14–4.19, p<0.001; aOR 2.98, 95% CI 2.00–4.44, p<0.001) (S5 Table).

## Discussion

This study examined the tobacco cessation following implementation of a cessation counseling intervention that was offered within routine care workflows to both smoked and smokeless tobacco users hospitalized in India. It adds to a very limited literature studying hospital-initiated cessation counseling support provided to individuals using smokeless tobacco. It also addresses a key gap in the literature, the implementation of hospital-initiated tobacco cessation interventions in LMIC settings, where interventions are often delivered by research nurses, and not studied in routine clinical practice [35]. Indeed, while the LifeFirst program is currently discontinued in the Prince Aly Khan Hospital during facilities construction, based on the experiences with LifeFirst described here, it is currently being implemented in six other Mumbai hospitals. While positive results have been demonstrated with medical student delivered counseling, solutions are needed to address tobacco among hospitalized patients in diverse hospital settings including private and other non-academic settings [36]. A cessation counseling program that can address both smoked and smokeless tobacco is especially important for India, other South Asian countries, and other regions that bear the greatest burden of morbidity and mortality from smokeless tobacco use [21].

The pre- and post-implementation samples in this study differed in attitudes and beliefs about tobacco and cessation, nicotine dependence and reported psychiatric symptoms. Post-implementation, a higher proportion of patients reported being somewhat or very motivated to quit, somewhat or very confident in their ability to quit and believing that tobacco has harmed their health compared to the pre-implementation period. Some of these changes over several years may be related to tobacco control activities being implemented in India during the study period including warning labels, mass media campaigns and product regulation [37]. Differences in nicotine dependence may be related to the evolving product landscape including product bans or changing patterns of use, as has been measured in other countries [38–40]. The decrease in symptoms of depression and anxiety is interestingly contradictory to overall trends in mental health in India [41]. The impacts of this difference in symptom burden is not clear. While mental health symptoms are associated with lower rates of quit success in high income settings, less is known about the relationship between mental health and cessation in LMIC settings [42]. In this case, PHQ-2 and GAD-2 scores were not associated with cessation in our adjusted models.

In the multivariable models, the only factor other than implementation phase that is independently associated with cessation in the adjusted model is a tobacco-related admission

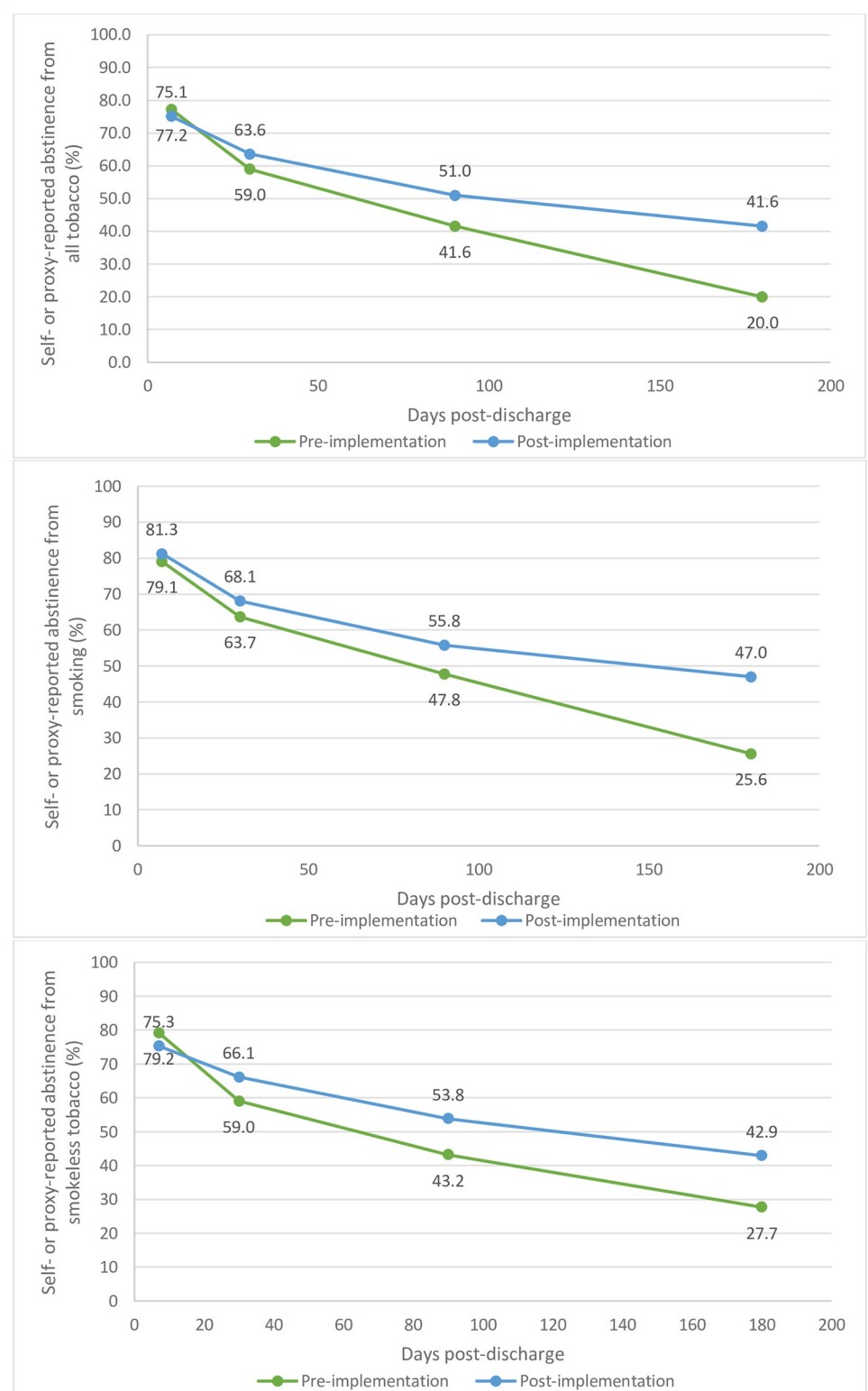

**Fig 2. Self- or proxy-reported continuous tobacco abstinence among baseline users by type and for all forms of tobacco at post-discharge follow-up (1 week [7 days], 1 month [30 days], 3 months [90 days] and 6 months [180 days]).**

**Table 2. Odds of continuous abstinence from all tobacco at 6 months comparing post-implementation vs. pre-implementation, n = 998.**

| | Odds Ratio | | | p-value | Adjusted Odds Ratio[a] | | | p-value |
|---|---|---|---|---|---|---|---|---|
| | OR | 95% LCL | 95% UCL | | aOR | 95% LCL | 95% UCL | |
| Post- vs. pre-implementation | 2.86 | 2.06 | 3.96 | <0.001 | 2.86 | 1.94 | 4.21 | <0.001 |

Abbreviations: OR = odds ratio, aOR = adjusted odds ratio, UCL = upper confidence limit, LCL = lower confidence limit, FTND-Fagerström test for nicotine dependence, PHQ-2 = patient health questionnaire 2 item instrument, GAD-2 = generalized anxiety disorder 2 item instrument.

[a]Adjusted for tobacco type, age, sex, education, employment, marital status, FTND (FTND for those who smoke only, FTND SL for those who use smokeless only or the highest of the two scores for dual users), confidence in quitting, motivation to quit, beliefs in harms of tobacco (Composite item of "Do you think tobacco use has harmed your health", "Is tobacco a cause of the illness you are in the hospital for?" and "Do you think quitting now would improve your health?" with responses categorized as Most: reported *Some* or *A lot* for all three items; Some: reported at least one *Some* or *A lot* in any of the three items; Little: reported *A little bit*, *Not at all*, or *Don't know/refused* for all three items), Admission diagnosis (tobacco related includes cardiovascular disease, chronic lung disease, or tobacco-related cancer), depression and anxiety symptoms.

diagnosis. This mirrors a prior analysis we conducted in this setting with patients using single product types, smoking or using smokeless tobacco, in which admission diagnosis predicted abstinence after discharge [43]. Given the impact of admission diagnosis on cessation, it is possible that programs designed to deliver disease-specific behavioral cessation support may yield further benefit [44].

It is also notable that the intervention consists of counseling alone. We measured self-reported use of evidence-based medication in follow-up and found it very rare pre- and post-implementation. Adding provision of pharmacotherapy to the counseling intervention should further enhance the effect [45].

## Limitations

Our pre-post study design has limitations. In this observational data we could adjust for several factors that changed from pre- to post-implementation including changes in attitudes, beliefs and changing tobacco use patterns over time. There may be other factors that we did not measure or adjust for that changed over time given that India has implemented tobacco control measures that may have contributed to the higher prevalence of cessation in the post-implementation phase. The pre-post design is susceptible to secular trends and imbalance between the pre- and post- groups by both measured and unmeasured confounders that could influence our results. This design allows us to measure associations but does not generate causal inference. The study design does not include a placebo or attention matched control in routine practice in the pre-implementation group. Loss to follow-up differences may bias our results. To minimize this potential source of bias, we used multiple imputation rather than categorizing those lost to follow-up as having continued tobacco use [34, 46]. We rely on self- or proxy-report of tobacco use rather than biochemical verification. While biochemical verification is often used in randomized trials, in this observational study of intervention delivery within routine hospital practices, it was not feasible [47]. Self- or proxy-report may be subject to social desirability bias and outcome assessors were aware of participant pre- or post-implementation status. Despite these limitations, our ability to adjust for a long list of potential confounders, our analytic designs including our intention to treat approach and use of multiple imputation to account for selection bias and differences in loss-to-follow-up, strive to minimize potential bias in this study.

## Conclusions

This study, conducted within routine care workflows in an LMIC clinical setting, provides information about tobacco behaviors after the implementation of LifeFirst, a hospital-initiated

cessation counseling program. Patients hospitalized after the program's implementation were substantially more likely to achieve tobacco abstinence than patients who were hospitalized before the program started. While temporal trends and differences in the pre- and post-implementation samples may contribute to this association in this observational design, the association we measured after adjusting for a long list of characteristics suggests that this evidence-based counseling intervention is a promising model for identifying and assisting people who use tobacco in a hospital setting that is worthy of further studies of its effectiveness and implementation. It aligns with Framework Convention on Tobacco Control Article 14, is feasible, and has the potential to improve health outcomes for hospitalized patients who use tobacco.

## Supporting information

**S1 File. Survey instrument.**
(PDF)

**S1 Table. LifeFirst protocol and contents by session.**
(DOCX)

**S2 Table. Tobacco product use in Pre- and Post-implementation samples.**
(DOCX)

**S3 Table. Outcome availability by follow-up timepoint and respondent.**
(DOCX)

**S4 Table. Sensitivity analyses.**
(DOCX)

**S5 Table. Odds of continuous abstinence from all tobacco at 6 months comparing all pre-implementation participants vs. those post-implementation participants who accepted LifeFirst counseling.**
(DOCX)

**S1 Data. Deidentified dataset.**
(XLSX)

## Acknowledgments

The team would like to thank Prince Aly Khan Hospital patients and staff for their contributions to this work. The team would like to thank the LifeFirst counsellors and data collectors–Jyoti Inamdar, Rashmi Asthana, Harshali Gaikwad and Deepali Patil. We would also like to thank Meghan Rieu-Werden for contributions to data curation for this analysis.

## Author Contributions

**Conceptualization:** Himanshu A. Gupte, Gina R. Kruse, Sultan Pradhan, Nancy A. Rigotti.

**Data curation:** Himanshu A. Gupte, Dinesh Jagiasi.

**Formal analysis:** Gina R. Kruse, Yuchiao Chang, Nancy A. Rigotti.

**Investigation:** Himanshu A. Gupte, Sultan Pradhan.

**Methodology:** Himanshu A. Gupte, Gina R. Kruse, Yuchiao Chang, Nancy A. Rigotti.

**Project administration:** Himanshu A. Gupte.

**Supervision:** Himanshu A. Gupte.

**Visualization:** Yuchiao Chang.

**Writing – original draft:** Himanshu A. Gupte, Gina R. Kruse, Nancy A. Rigotti.

**Writing – review & editing:** Himanshu A. Gupte, Gina R. Kruse, Dinesh Jagiasi, Sultan Pradhan, Nancy A. Rigotti.

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
