## [Decision Letter · Decision Letter 0]

18 Jun 2024

PONE-D-24-16725Effectiveness of tobacco cessation counseling to promote post-discharge tobacco abstinence in a Mumbai hospital: A pragmatic evaluation of the LifeFirst programPLOS ONE

Dear Dr. Kruse,

Thank you for submitting your manuscript to PLOS ONE. After careful consideration, we feel that it has merit but does not fully meet PLOS ONE’s publication criteria as it currently stands. Therefore, we invite you to submit a revised version of the manuscript that addresses the points raised during the review process.

We look forward to receiving your revised manuscript.

Kind regards,

Palash Chandra Banik, MPhil

Academic Editor

PLOS ONE

Journal Requirements:

2. Thank you for stating the following in the Competing Interests section: "GK has a family financial interest in Dimagi Inc. She was not involved in selection of the data collection platform. GK has received research grants from the National Comprehensive Cancer Network with funding from Astra Zeneca. HG is an employee of the Narotam Sekhsaria Foundation. All other authors declare that they have no conflicts of interest." 

3. In the online submission form, you indicated that the datasets generated and/or analysed during the current study are not publicly available due to local institutional standards, but are available from the corresponding author on reasonable request. De-identified data from this study will be made available (as allowable according to institutional IRB standards) by emailing the corresponding author. Analytic code used to conduct the analyses presented in this study are not available in a public archive. They may be available by emailing the corresponding author. Materials used to conduct the study are not publicly available.

Reviewers' comments:

Reviewer's Responses to Questions

**Comments to the Author**

1. Is the manuscript technically sound, and do the data support the conclusions?

Reviewer #1: Partly

2. Has the statistical analysis been performed appropriately and rigorously? 

Reviewer #1: Yes

3. Have the authors made all data underlying the findings in their manuscript fully available?

Reviewer #1: Yes

4. Is the manuscript presented in an intelligible fashion and written in standard English?

Reviewer #1: Yes

5. Review Comments to the Author

Reviewer #1: The authors report the effectiveness of a tobacco counselling program (from hospitalization to 6 months post discharge) in a single hospital in Mumbai (India), using a pre-post study design. They found the intervention effective, with an OR of 2.95.

Strengths of the study include a focus on all forms of tobacco (including smokeless tobacco), a comprehensive intervention with structured counsellor training, and an effort to apply it hospital wide.

The study has several methodological limitations that constrain causal inferences. First, the primary outcome is self-reported, rather than a biochemical measure. Second, the pre-post study design is prone to influence from secular trends and unbalanced covariates (seen in Table 1, for instance in confidence in quitting smoking). In addition, outcome assessors were therefore not blinded to study arm assignment, which may have influenced the results. Third, the post period has more patient contacts (duration, if not frequency) than the pre-period, which makes the study susceptible to the Hawthorne effect. This is particularly important given the self-reported nature of the primary outcome. Fourth, 6 month follow up was only obtained in 50% of patients in the pre-group, and 70% in the post group. The overall rates are low, with wide differences between the two groups. Due to these factors, the OR obtained by the authors, is more likely than not, an overestimate of the causal effect of the intervention. The study should be reported as an observational study, given these limitations, and no reliable causal inferences can be made with regards to the effect of the intervention.

Abstract

Conclusion needs to be reworded, to emphasize limitations of study design. Please remove phrases such as ‘real world evidence’. No causal inference can be made from this study, only associations of unclear causality.

Results

Table 2- I am not quite sure what is this table trying to show. There is the primary outcome reported in the first row, and the subsequent rows show the effect of other variables (such as employment/education) on tobacco cessation rates. This should be in a separate table with its own heading, or the current heading should be amended. The current heading is misleading.

Lines 260-262- ‘Follow-up assessments were completed more frequently in the post-implementation phase than the pre-implementation phase (Supplemental Table C, differences all p<0.001).’ Please provide the actual numbers/proportions in the results (the 6 month follow up rates, 50% vs 70%), and not just the p value. This is a crucial piece of information, and should be in the main text.

Discussion

‘It is also one of few studies to assess the feasibility and effectiveness of a hospital-initiated

tobacco cessation intervention conducted outside a high-income country. Whether the

effectiveness of hospital-based cessation treatment that has been demonstrated in high-income settings translates to LMIC settings is not yet established.’

This is not true, based on the references from the introduction of this very paper. An additional reference is below,

https://link.springer.com/article/10.1007/s11606-023-08243-y

Conclusions

Lines 358-260: ‘…the magnitude of the difference after adjusting for a long list of measured

characteristics supports the conclusion that this evidence-based counseling intervention

benefits patients and increases abstinence from all tobacco.’ Extremely flawed methods will provide extremely flawed results. The magnitude of the difference, if anything, is likely a result of the flawed study design. Please take a cautious approach to interpreting the observational data, rather than overemphasizing causal inferences (which really cannot be made).

Other points

‘Real world effectiveness’- Please refrain from using this term, as there is no standard definition, and there is only one world, which is real. Using this term distracts from the methodological weaknesses of the study, which the authors should be more upfront about.

6. PLOS authors have the option to publish the peer review history of their article (what does this mean?). If published, this will include your full peer review and any attached files.

Reviewer #1: **Yes: **Aditya Khetan

---

## [Author Response · Author response to Decision Letter 0]

29 Jul 2024

Response to reviews:

1. The study has several methodological limitations that constrain causal inferences. First, the primary outcome is self-reported, rather than a biochemical measure. 

We have reworded this limitation for emphasis.

“While biochemical verification is common in randomized trials, in this observational study of intervention delivery within routine hospital practices, it was not feasible.”

2. Second, the pre-post study design is prone to influence from secular trends and unbalanced covariates (seen in Table 1, for instance in confidence in quitting smoking). 

We emphasize this limitation of our observational design. 

”The pre-post design is susceptible to secular trends and imbalance between the pre- and post- groups by both measured and unmeasured confounders that could influence our results.”

3. In addition, outcome assessors were therefore not blinded to study arm assignment, which may have influenced the results. 

Patients were not assigned to study arms in this observational study, but we have added a comment to limitations noting that outcome assessors were aware of pre- or post-group status.

“Self- or proxy-report may be subject to social desirability bias and outcome assessors were aware of participant pre- or post-implementation status.”

4. Third, the post period has more patient contacts (duration, if not frequency) than the pre-period, which makes the study susceptible to the Hawthorne effect. This is particularly important given the self-reported nature of the primary outcome. 

We have added this point to our limitations.

“The study design does not include a placebo or attention matched control in routine practice in the pre-implementation group.”

5. Fourth, 6 month follow up was only obtained in 50% of patients in the pre-group, and 70% in the post group. The overall rates are low, with wide differences between the two groups. Due to these factors, the OR obtained by the authors, is more likely than not, an overestimate of the causal effect of the intervention. The study should be reported as an observational study, given these limitations, and no reliable causal inferences can be made with regards to the effect of the intervention.

We agree with this point. It is stated in our “Transparency and registration” section where we noted this is observational, and noted it as well in other places in the manuscript. To further address this reviewer comment and eliminate any implication that we are inferring causality we edited the title and an explicit statement about causal inference.

“This design allows us to measure associations, but does not generate causal inference.”

6. Conclusion needs to be reworded, to emphasize limitations of study design. Please remove phrases such as ‘real world evidence’. No causal inference can be made from this study, only associations of unclear causality.

We have edited the Abstract conclusions, removed the term ‘real-world’ and reworded to clarify that our study generated measures of association only.

“Post-discharge abstinence was more common after implementation of a multi-session tobacco counseling program for hospitalized patients compared to abstinence among patients hospitalized before implementation. These findings represent observational evidence of a promising association between post-discharge abstinence and a hospital-based tobacco cessation program implemented in routine practice in an LMIC setting.”

7. Results; Table 2- I am not quite sure what is this table trying to show. There is the primary outcome reported in the first row, and the subsequent rows show the effect of other variables (such as employment/education) on tobacco cessation rates. This should be in a separate table with its own heading, or the current heading should be amended. The current heading is misleading.

We have edited this to table to present the primary outcome. We have edited the footnote to list all of the variables adjusted for in the adjusted model and edited the table title.

8. Lines 260-262- ‘Follow-up assessments were completed more frequently in the post-implementation phase than the pre-implementation phase (Supplemental Table C, differences all p<0.001).’ Please provide the actual numbers/proportions in the results (the 6 month follow up rates, 50% vs 70%), and not just the p value. This is a crucial piece of information, and should be in the main text.

We have edited the text and added this information to the main text.

“The proportion of assessments completed by self or proxy was higher post-implementation at all follow-up timepoints (S3 Table; 1 week: pre 62.5% vs. post 83.1%; 1 month: pre 58.1% vs. post 80.2%; 3 months: pre 51.7% vs. post 70.6%, 6 months: pre 49.7% vs. post 70.4%, differences all p<0.001).”

9. Discussion; ‘It is also one of few studies to assess the feasibility and effectiveness of a hospital-initiated tobacco cessation intervention conducted outside a high-income country. Whether the effectiveness of hospital-based cessation treatment that has been demonstrated in high-income settings translates to LMIC settings is not yet established.’

This is not true, based on the references from the introduction of this very paper. An additional reference is below, https://link.springer.com/article/10.1007/s11606-023-08243-y

We have edited this statement and cited the suggested reference.

“It adds to a very limited literature studying hospital-initiated cessation counseling support provided to individuals using smokeless tobacco. It also addresses a key gap in the literature, the implementation of hospital-initiated tobacco cessation interventions in LMIC settings, where interventions are often delivered by research nurses, and not studied in routine clinical practice. Positive results have been demonstrated with medical student delivered counseling. Solutions are needed to address tobacco among hospitalized patients in diverse hospital settings including private and other non-academic settings.”

10. Conclusions; Lines 358-260: ‘…the magnitude of the difference after adjusting for a long list of measured characteristics supports the conclusion that this evidence-based counseling intervention benefits patients and increases abstinence from all tobacco.’ Extremely flawed methods will provide extremely flawed results. The magnitude of the difference, if anything, is likely a result of the flawed study design. Please take a cautious approach to interpreting the observational data, rather than overemphasizing causal inferences (which really cannot be made).

We worked to better emphasize the many limitations in this observational pre-post study design pointed out by the reviewer. However, we were not attempting to present this as a ‘flawed’ RCT. It is a different study design that allows for measures of association within routine practice workflows rather than a controlled experimental trial context. We edited the manuscript to ensure no claim of causality is included in the text and continue to repeatedly label it as observational. We also reworded the sentence noted above.

“While temporal trends and differences in the pre- and post-implementation samples may contribute to this association in this observational design, the association we measured after adjusting for a long list of characteristics suggests that this evidence-based counseling intervention is a promising model for identifying and assisting people who use tobacco in a hospital setting that is worthy of further studies of its effectiveness and implementation.”

11. Other points; ‘Real world effectiveness’- Please refrain from using this term, as there is no standard definition, and there is only one world, which is real. Using this term distracts from the methodological weaknesses of the study, which the authors should be more upfront about.

We have removed this term. We have edited the text to expand upon the limitations of this observational study design. We continue to label this study as observational, as we did in the original submission. While we did not explicitly describe a causal association in our original discussion, in response to the reviewer’s feedback we adjusted our wording so as not to not imply causality. We believe the edits prompted by this feedback have improved the clarity of our conclusions. This was not intended to distract. We do think there are strengths to our observational design compared to RCTs. RCTs are experimental, they require study workflows beyond routine clinical practice (e.g. randomization, placebo or attention matched controls) and these factors contribute to challenges in translating promising effects from RCTs into routine clinical practice. Observational studies in routine practice have inherent value, not as ‘flawed’ RCTs but as their own contribution to the literature. 

12. Journal Requirements: When submitting your revision, we need you to address these additional requirements. Please ensure that your manuscript meets PLOS ONE's style requirements, including those for file naming. The PLOS ONE style templates can be found at https://journals.plos.org/plosone/s/file?id=wjVg/PLOSOne_formatting_sample_main_body.pdf and https://journals.plos.org/plosone/s/file?id=ba62/PLOSOne_formatting_sample_title_authors_affiliations.pdf

We reviewed style requirements and made edits to file naming.

13. Thank you for stating the following in the Competing Interests section: "GK has a family financial interest in Dimagi Inc. She was not involved in selection of the data collection platform. GK has received research grants from the National Comprehensive Cancer Network with funding from Astra Zeneca. HG is an employee of the Narotam Sekhsaria Foundation. All other authors declare that they have no conflicts of interest." 

We have added the updated Competing Interests statement with the detail above to our cover letter.

14. In the online submission form, you indicated that the datasets generated and/or analysed during the current study are not publicly available due to local institutional standards, but are available from the corresponding author on reasonable request. De-identified data from this study will be made available (as allowable according to institutional IRB standards) by emailing the corresponding author. Analytic code used to conduct the analyses presented in this study are not available in a public archive. They may be available by emailing the corresponding author. Materials used to conduct the study are not publicly available.

We have generated a deidentified dataset that is in compliance with ethical requirements and uploaded it with our resubmission.

15. Your ethics statement should only appear in the Methods section of your manuscript. If your ethics statement is written in any section besides the Methods, please move it to the Methods section and delete it from any other section. Please ensure that your ethics statement is included in your manuscript, as the ethics statement entered into the online submission form will not be published alongside your manuscript. 

We have ensured the ethics statement is included only in the Methods section.

16. Please include captions for your Supporting Information files at the end of your manuscript, and update any in-text citations to match accordingly. Please see our Supporting Information guidelines for more information: http://journals.plos.org/plosone/s/supporting-information. 

We have added captions and in-text citations using the style noted in the links above. 

We reviewed our reference list.

---

## [Decision Letter · Decision Letter 1]

16 Aug 2024

PONE-D-24-16725R1Post-discharge tobacco abstinence in a Mumbai hospital after implementation of tobacco cessation counseling: A pragmatic evaluation of the LifeFirst programPLOS ONE

Dear Dr. Kruse,

Thank you for submitting your manuscript to PLOS ONE. After careful consideration, we feel that it has merit but does not fully meet PLOS ONE’s publication criteria as it currently stands. Therefore, we invite you to submit a revised version of the manuscript that addresses the points raised during the review process.

We look forward to receiving your revised manuscript.

Kind regards,

Palash Chandra Banik, MPhil

Academic Editor

PLOS ONE

Journal Requirements:

Reviewers' comments:

Reviewer's Responses to Questions

**Comments to the Author**

1. If the authors have adequately addressed your comments raised in a previous round of review and you feel that this manuscript is now acceptable for publication, you may indicate that here to bypass the “Comments to the Author” section, enter your conflict of interest statement in the “Confidential to Editor” section, and submit your "Accept" recommendation.

Reviewer #1: All comments have been addressed

Reviewer #2: (No Response)

2. Is the manuscript technically sound, and do the data support the conclusions?

Reviewer #1: Yes

Reviewer #2: Yes

3. Has the statistical analysis been performed appropriately and rigorously? 

Reviewer #1: Yes

Reviewer #2: I Don't Know

4. Have the authors made all data underlying the findings in their manuscript fully available?

Reviewer #1: Yes

Reviewer #2: Yes

5. Is the manuscript presented in an intelligible fashion and written in standard English?

Reviewer #1: Yes

Reviewer #2: Yes

6. Review Comments to the Author

Reviewer #1: All comments have been addressed. No further concerns. I think it is suitable for publication.

All comments have been addressed. No further concerns. I think it is suitable for publication.

Reviewer #2: This pre-post study measures tobacco abstinence (including smokeless tobacco) among individuals discharged from a Mumbai hospital after the implementation of cessation counseling (LifeFirst) compared to abstinence among those discharged pre-implementation. Given that most tobacco-related diseases occur in low- and middle-income countries, this study addresses a critical need. In addition, few studies have assessed smoking cessation treatment initiated in the hospital in low- and- middle income settings.

• I would like to know more about LifeFirst Services. Are these services still available at the hospital? LifeFirst does not include recommendations for pharmacotherapy. Is this because of availability?

• The authors have addressed the many limitations inherent in observational studies. In future studies, other designs, such as regression discontinuity or difference-in-difference (DD) approaches could be considered to address some of these limitations.

• The authors can consider citing the reference below which discusses that for some study designs, such as Quitline studies and observational studies, biochemical validation may not be feasible (Reviewer’s comment #1)

Benowitz NL, Bernert JT, Foulds J, Hecht SS, Jacob P, Jarvis MJ, Joseph A, Oncken C, Piper ME. Biochemical Verification of Tobacco Use and Abstinence: 2019 Update. Nicotine Tob Res. 2020 Jun 12;22(7):1086-1097. doi: 10.1093/ntr/ntz132. PMID: 31570931; PMCID: PMC7882145.

7. PLOS authors have the option to publish the peer review history of their article (what does this mean?). If published, this will include your full peer review and any attached files.

Reviewer #1: **Yes: **Aditya Khetan

Reviewer #2: No

---

## [Author Response · Author response to Decision Letter 1]

3 Sep 2024

Response to review:

Journal: Please review your reference list to ensure that it is complete and correct. If you have cited papers that have been retracted, please include the rationale for doing so in the manuscript text, or remove these references and replace them with relevant current references. Any changes to the reference list should be mentioned in the rebuttal letter that accompanies your revised manuscript. If you need to cite a retracted article, indicate the article’s retracted status in the References list and also include a citation and full reference for the retraction notice.

We pulled up all cited articles in PubMed and found none of our cited works have been retracted. 

Reviewers' comments:

Reviewer #2: This pre-post study measures tobacco abstinence (including smokeless tobacco) among individuals discharged from a Mumbai hospital after the implementation of cessation counseling (LifeFirst) compared to abstinence among those discharged pre-implementation. Given that most tobacco-related diseases occur in low- and middle-income countries, this study addresses a critical need. In addition, few studies have assessed smoking cessation treatment initiated in the hospital in low- and- middle income settings.

• I would like to know more about LifeFirst Services. Are these services still available at the hospital? LifeFirst does not include recommendations for pharmacotherapy. Is this because of availability?

We have added additional detail about LifeFirst. Pharmacotherapy was not an integral part of LifeFirst because of limited evidence supporting pharmacotherapy use in smokeless tobacco and the high cost of NRT and prescription pharmacotherapies. The program is currently discontinued in the hospital while it undergoes construction. However, based on the experience described in this study the service was expanded and the model is currently being implemented in six other hospitals in Mumbai. 

LifeFirst Intervention

“Pharmacotherapy is not integral to the program due to the high costs of nicotine replacement therapy and prescription pharmacotherapies and because of the large proportion of patients using smokeless only, for whom there is less evidence supporting pharmacotherapy use.”

Discussion

“Adding provision of pharmacotherapy to the counseling intervention should further enhance the effect.”

and

“Indeed, while the LifeFirst program is currently discontinued in the Prince Aly Khan Hospital during facilities construction, based on the experiences with LifeFirst described here, it is currently being implemented in six other Mumbai hospitals.”

• The authors have addressed the many limitations inherent in observational studies. In future studies, other designs, such as regression discontinuity or difference-in-difference (DD) approaches could be considered to address some of these limitations.

Thank you for this suggestion, we will consider this in future studies where an observational design is required.

• The authors can consider citing the reference below which discusses that for some study designs, such as Quitline studies and observational studies, biochemical validation may not be feasible (Reviewer’s comment #1)

Benowitz NL, Bernert JT, Foulds J, Hecht SS, Jacob P, Jarvis MJ, Joseph A, Oncken C, Piper ME. Biochemical Verification of Tobacco Use and Abstinence: 2019 Update. Nicotine Tob Res. 2020 Jun 12;22(7):1086-1097. doi: 10.1093/ntr/ntz132. PMID: 31570931; PMCID: PMC7882145.

We have added reference to this study.

---

## [Decision Letter · Decision Letter 2]

8 Sep 2024

Post-discharge tobacco abstinence in a Mumbai hospital after implementation of tobacco cessation counseling: A pragmatic evaluation of the LifeFirst program

PONE-D-24-16725R2

Dear Dr. Kruse,

We’re pleased to inform you that your manuscript has been judged scientifically suitable for publication and will be formally accepted for publication once it meets all outstanding technical requirements.

Kind regards,

Palash Chandra Banik, MPhil

Academic Editor

PLOS ONE

Additional Editor Comments (optional):

Reviewers' comments:

Reviewer's Responses to Questions

**Comments to the Author**

1. If the authors have adequately addressed your comments raised in a previous round of review and you feel that this manuscript is now acceptable for publication, you may indicate that here to bypass the “Comments to the Author” section, enter your conflict of interest statement in the “Confidential to Editor” section, and submit your "Accept" recommendation.

Reviewer #2: All comments have been addressed

2. Is the manuscript technically sound, and do the data support the conclusions?

Reviewer #2: Yes

3. Has the statistical analysis been performed appropriately and rigorously? 

Reviewer #2: Yes

4. Have the authors made all data underlying the findings in their manuscript fully available?

Reviewer #2: Yes

5. Is the manuscript presented in an intelligible fashion and written in standard English?

Reviewer #2: Yes

6. Review Comments to the Author

Reviewer #2: Thank you for addressing all of the feedback. I have no further suggestions or feedback.

7. PLOS authors have the option to publish the peer review history of their article (what does this mean?). If published, this will include your full peer review and any attached files.

Reviewer #2: No

---

## [Editor Report · Acceptance letter]

15 Oct 2024

PONE-D-24-16725R2 

PLOS ONE

Dear Dr. Kruse, 

I'm pleased to inform you that your manuscript has been deemed suitable for publication in PLOS ONE. Congratulations! Your manuscript is now being handed over to our production team.

Kind regards, 

on behalf of

Dr. Palash Chandra Banik 

Academic Editor

PLOS ONE